# Senopathies—Diseases Associated with Cellular Senescence

**DOI:** 10.3390/biom13060966

**Published:** 2023-06-08

**Authors:** Oleh Lushchak, Markus Schosserer, Johannes Grillari

**Affiliations:** 1Ludwig Boltzmann Institute for Traumatology, The Research Center in Cooperation with AUVA, 1200 Vienna, Austria; oleh.lushchak@pnu.edu.ua; 2Department of Biochemistry and Biotechnology, Precarpathian National University, 76000 Ivano-Frankivsk, Ukraine; 3Research and Development University, 76018 Ivano-Frankivsk, Ukraine; 4Institute of Medical Genetics, Center for Pathobiochemistry and Genetics, Medical University of Vienna, 1090 Vienna, Austria; markus.schosserer@meduniwien.ac.at; 5Austrian Cluster for Tissue Regeneration, 1200 Vienna, Austria; 6Institute of Molecular Biotechnology, Department of Biotechnology, University of Natural Resources and Life Sciences, 1190 Vienna, Austria

**Keywords:** cellular senescence, aging, senopathy, senolytic, senomorphic, senotherapy, geroscience, senescence-associated secretory phenotype (SASP), pathology, age-related disease

## Abstract

Cellular senescence describes a stable cell cycle arrest state with a characteristic phenotype. Senescent cells accumulate in the human body during normal aging, limiting the lifespan and promoting aging-related, but also several non-related, pathologies. We propose to refer to all diseases whose pathogenesis or progression is associated with cellular senescence as “senopathies”. Targeting senescent cells with senolytics or senomorphics is likely to mitigate these pathologies. Examples of senopathies include cardiovascular, metabolic, musculoskeletal, liver, kidney, and lung diseases and neurodegeneration. For all these pathologies, animal studies provide clear mechanistic evidence for a connection between senescent cell accumulation and disease progression. The major persisting challenge in developing novel senotherapies is the heterogeneity of senescence phenotypes, causing a lack of universal biomarkers and difficulties in discriminating senescent from non-senescent cells.

## 1. Introduction

Cellular senescence is a state of stable cell cycle arrest. It is, in vivo, most commonly induced in response to various physiological and pathological conditions through DNA damage signaling [1,2,3,4] or the expression of oncogenes [5,6]. Leonard Hayflick discovered cellular senescence in 1965 as a phenomenon when human fetal fibroblasts stopped dividing but maintained viability and metabolic activity after prolonged cultivation [7]. In these early days, the permanent cell cycle arrest and characteristic morphological changes associated with senescence were considered mere artifacts of in vitro cultivation of primary cells [8,9]. Several decades later, the progressive shortening of telomeres was shown to be associated with the proliferation limit of cells in culture, known now as replicative senescence [10]. After that, an increasing body of evidence emerged, demonstrating that senescent cells (SCs) also accumulate in vivo, preferentially in aged individuals, and specifically at sites of aging-related pathologies [11,12,13]. Ultimately, elegant experiments in mice showed that the accumulation of SCs during aging limits the healthy murine lifespan [14,15]. These discoveries initiated ongoing endeavors to identify novel “senotherapies”, which comprise pharmacological compounds to eliminate SCs specifically (“senolytics”) or to mitigate or modify their senescence-associated secretory phenotype (SASP) (“senomorphics”) [16,17,18,19,20,21].

The SASP is considered the main trigger of tissue damage caused by SCs and is composed of pro-inflammatory cytokines, chemokines, and matrix-remodeling factors [22]. In addition, extracellular vesicles, miRNAs [23,24,25,26], and pro-inflammatory lipids [27] have been identified as SASP members. Recent evidence suggests that SCs attract innate immune cells by their SASP and activate dendritic and CD8 T-cells by increased MHC class I expression and self-antigen presentation [28]. However, SCs are also highly efficient in resisting immune clearance, partially explaining the increased SC burden in aged individuals. Cellular senescence is also associated with increased cell plasticity and a concomitant loss of cell identity. Senescent cells promote the de-differentiation of non-senescent neighboring cells, causing further tissue damage [29]. This relatively new concept is supported by a study demonstrating that Alzheimer’s disease (AD) patients induced neurons simultaneously expressed markers of de-differentiation and cellular senescence [30]. On the contrary, SCs play essential roles in embryogenesis, wound healing, and tissue regeneration [31,32]. Thus, while the transient presence of SCs is beneficial for maintaining tissue homeostasis and preventing tumorigenesis early in life, the accumulation of these cells drives age-related disease and promotes chronic inflammation [31].

Similarly, the contribution of cellular senescence to a wide range of pathological conditions is highly complex, combining beneficial and detrimental effects [1,33]. Many, but not all, of these are associated with aging, such as cardiovascular pathologies [12,34], neurodegeneration [35], and ocular diseases [36]. However, it is often unclear if the accumulation of SCs is a cause or consequence of the respective pathological condition [37]. Cystic fibrosis [38] and acute infections, such as SARS-CoV-2 [39], are not considered age-related pathologies but are both characterized and worsened by SC presence and accumulation.

We propose to refer to all diseases whose pathogenesis or progression is associated with cellular senescence as “senopathies” and aim to provide examples and mechanistic explanations in this opinion paper. While, on the one hand, precision medicine identifies more and more subtypes of disease entities, senescence might, on the other hand, be a common denominator and therapeutic (co-) target for a variety of very different disease entities. Here, we do not cover the connection between cellular senescence, cancer, and senolytic cancer therapy. This topic has already been extensively reviewed elsewhere [40,41,42] and is recently extended by the idea of a one-two-punch strategy against cancer [43].

## 2. Biomarkers of Cellular Senescence

In 1995, Dimri and colleagues demonstrated that cellular senescence occurs in cultures of human cells and aging skin by developing a first biomarker, such as the activity of β-galactosidase at pH 6.0, referred to as SA-β-Gal [11]. Increased SA-β-Gal due to elevated lysosomal content with altered activity is widely adopted as a biomarker both in vitro and in vivo. Later studies showed that SCs are distinct from quiescent and differentiated cells with specific morphological and metabolic signatures, reorganized chromatin, altered gene expression, and expression of the SASP. Before highlighting biomarkers of SCs based on these characteristic features, it is essential to emphasize that not all senescent cells display the full panel of all senescence biomarkers. Moreover, some senescence markers are also present in apoptotic and quiescent cells. Thus, only complex phenotyping using several biomarkers might lead to correct conclusions.

In general, SCs are characterized by a stable cell cycle arrest primarily due to DNA damage signaling in response to the shortening of telomeres, exposure to growth factors and inflammatory cytokines, high levels of reactive oxygen species (ROS), and mitochondrial dysfunction. These cells are typically larger, displaying a flattened shape and extensive vacuolization. Disrupted integrity of the nuclear envelope is observed due to a loss of Lamin B1 protein, whose decreased expression is currently one of the most useful senescence biomarkers [44]. Senescent cells accumulate many dysfunctional mitochondria producing increased levels of ROS, further causing DNA damage [45].

Chromatin reorganization in SCs provokes the formation of senescence-associated heterochromatin foci (SAHFs) to induce the silencing of genes that promote proliferation, including E2F target genes. SAHFs are characterized by extensive DAPI staining and immunoreactivity to macroH2A, heterochromatin protein 1 (HP1), and lysine 9 di-or-tri-methylated histone H3 (H3K9me2/3) [46].

Moreover, SCs display a persistent DNA damage response (DDR) and contain nuclear foci called “DNA segments with chromatin alterations reinforcing senescence” (DNA-SCARS). DNA-SCARS in uncapped telomeres are called “telomere dysfunction-induced foci” (TIF). Another indicator of the DDR is the phosphorylated form of H2A.X or γ-H2A.X, a variant of a histone required for checkpoint-mediated cell cycle arrest and DNA repair of double-stranded DNA breaks. Regulatory protein p53, as a mediator of the DDR, promotes the activation of p21^WAF1/CIP1^ (CDKN1A) and p16^INK4A^ (CDKN2A) to inhibit the activity of cyclin-dependent kinase [3]. Both protein and mRNA levels for p53, p16^INK4A^, and p21^WAF1/CIP1^, are extensively used to detect cellular senescence in vitro and in vivo [3,37,47].

Senescent cells maintain their metabolic activity and secrete inflammatory cytokines (IL-6, IL-8, TNF-⍺), growth factors such as TGFβ, chemokines, specific micro-RNAs, and matrix metalloproteinases, collectively known as the SASP [47]. The SASP modulates the surrounding environment, exerting its pathophysiological effects [47,48]. Given the nature of the SASP, senescence signaling is often linked to inflammation. Remarkably, some SASP factors, such as colony-stimulating factor 1 (CSF1), CCL2, and IL-8, recruit the immune system to promote the self-clearance of SCs [4]. This process seems critical to control the levels of SCs in a given setting and prevent chronic inflammatory responses.

Manifestation of each senescence hallmark is context-dependent and varies according to the trigger, cell or tissue type, and time from induction of the senescence program [49]. Changes at the levels of transcripts or proteins in response to multiple senescence inducers in different cell types demonstrate variability within specific pathological contexts and the requirement for a multi-marker system to identify SCs with precise accuracy [50]. A three-step multi-marker workflow was suggested to include (1) SA-β-gal or lipofuscin accumulation; (2) co-staining with markers frequently present (p16^INK4A^, p21^WAF1/CIP1^) or absent (proliferation markers, Lamin B1); and (3) identification of factors predicted to be altered in specific senescence contexts (SASP, DNA damage, and PI3K/FOXO/mTOR) [3]. Similarly, recent endeavors led to the development of the SenMayo marker panel, which identifies SCs based on their mRNA expression signatures across different tissues with high accuracy [51].

## 3. Selected Examples of Senopathies and Their Treatment with Senotherapies

Aging and related functional declines are strong drivers for many diseases and can be treated pharmacologically [52,53,54,55]. Aging-related senopathies are characterized by the accumulation of SCs over time, causing tissue and organ dysfunctions and the development of pathophysiological changes [56,57]. Targeting SCs with senolytics; however, may improve the functionality of various organ systems and prevent disease progression [18,20,58,59,60]. Similar mechanistic relationships are more difficult to establish for senomorphics because these compounds target the SASP, not SCs as such. Since most SASP components are commonly present in inflammatory diseases, it is difficult to determine if a therapeutically effective compound’s mode of action is via its senomorphic or more general anti-inflammatory properties. Thus, readouts, such as the most common SASP factors IL-6 or IL-8, are insufficient to establish a clear mechanistic connection between SCs and a specific disease. Several compounds with senomorphic activity in vitro, such as rapamycin, metformin, aspirin, statins and NF-κB-, p38MAPK-, JAK/STAT-, and ATM-inhibitors, are effective in potential senopathies [61].

Several studies connecting SC accumulation to disease progression used the *p16-3MR* [32] or the *INK-ATTAC* [14] mouse model. Both models contain a transgene that permits the visualization and inducible elimination of p16^INK4A^-expressing cells.

### 3.1. Cardiovascular Diseases

Improved cardiovascular function, increased regeneration, and prevention of hypertrophy and fibrosis were shown under the treatment of old animals with navitoclax or a combination of dasatinib and quercetin (D+Q) [16,62]. These drugs significantly decreased the SC burden, shown by reduced SA-β-gal staining, telomere-associated DNA damage foci (TAF), and p16^INK4A^ mRNA. Navitoclax effectively eliminated SCs from advanced atherosclerotic lesions in LDL receptor knockout mice exposed to a high-fat diet. Thereby, navitoclax prevented elastic fiber degradation and fibrous cap thinning through increased metalloprotease production [63].

### 3.2. Metabolic Disorders

Ruxolitinib treatment enhanced adipogenesis, reduced fat tissue loss and lipotoxicity, and improved insulin sensitivity of old animals by reducing the amounts of SA-β-gal positive cells and transcription of IL-6, p21^WAF1/CIP1^, and p16^INK4A^ [64]. Even more so, diabetes was prevented in the *NOD* mouse model of type 1 diabetes, pointing toward diabetes as a potential senopathy [65]. Alleviated anxiety-related behavior and decreased inflammation in *db*/*db* diabetes-prone animals fed a high-fat diet were observed after treatment with D+Q [66]. A subset of insulin-producing pancreatic β-cells acquires a SASP during the natural history of T1D in humans. Senescent β-cells upregulated the pro-survival mediator Bcl-2 in non-obese diabetic mice. Thompson and colleagues used ABT-737 and ABT-199, inhibitors of Bcl-2, to eliminate senescent β-cells sufficiently to prevent diabetes [65]. Metformin is an approved drug for type 2 diabetes in humans, but also positively affects other age-related disorders [67] and extends the healthy lifespan of male mice [68]. Although several senescence markers in different cell types are down-regulated upon exposure to this drug [69,70], the precise mechanisms of metformin action remain elusive [61].

### 3.3. Musculoskeletal Diseases

Significant reduction in SCs and SASP factors in old mice by treatment with Ruxolitinib or D+Q prevented loss of bone mass and increased their strength, effectively preventing osteoporosis and frailty [71]. Moreover, UBX0101 inhibited erosion of articular cartilage by targeting transcription of SASP-associated genes [72] and thus points towards osteoarthritis as senopathy as well. Aspirin, for which senomorphic properties were described [73], is also effective in arthritis and osteoporosis [74]. The novel small molecule NF-κB-inhibitor SR12343 reduced senescence markers and improved muscle pathologies in the *Zmpste24^−/−^* progeroid mouse model [75].

### 3.4. Liver and Kidney Diseases

Age-related liver steatosis and fibrosis were effectively treated with A-1331852 or D+Q by elimination of SCs with decreased production of the SASP, both in *INK-ATTAC* and in the *Mdr2^−/−^* mouse model of primary sclerosing cholangitis [76,77]. Moreover, treatment with FOXO4-DRI improved kidney function by affecting amounts of Lamin B1 positive cells and expression of IL-6 [21].

### 3.5. Neurodegeneration

Senopathies have detrimental effects on the nervous system by affecting both its structure and function. One prominent example of a neurodegenerative condition likely promoted by the accumulation of SCs and characterized by the expression of tau is AD. Studies in *SAMP8* mice developing pathophysiological alterations similar to human AD and in the *tau_NFT_-Mapt^0/0^* model acquiring tau-related pathologies at a faster rate show that treatments with fisetin or D+Q decreased inflammation and stress with restored synaptic function, cognitive deficits, and cerebral blood flow [78,79]. Moreover, navitoclax prevented cognitive function decline, gliosis, and hyper-phosphorylation of tau in *INK-ATTAC PS19* mice, which express high levels of human tau, specifically in neurons [80].

### 3.6. Lung Diseases

An accumulation of senescent cells induces fibrotic pulmonary disease, especially in idiopathic pulmonary fibrosis [81]. D+Q improved lung function in bleomycin-treated mice by affecting the mRNA levels of Mcpl, 11-6, and Mmpl2 [81]. Moreover, signatures of pulmonary fibrosis induced by irradiation were successfully treated with navitoclax [82]. In this model, the elimination of SCs and decreased inflammation were shown by reduced amounts of SA-β-gal positive cells and mRNA levels of p16^INK4A^, Bcl-2, IL-la, and IL-1β. It would be interesting if also chronic obstructive pulmonary disease (COPD), in which senescent cells are known to accumulate, were alleviated by senolytic treatment [83].

In Table 1, we summarize the effects of senolytic treatment on cardiovascular diseases, detrimental changes in bones, cartilage, liver, kidney, brain, and fat tissues, as well as treatment of atherosclerosis, neurodegeneration, fibrotic pulmonary disease, and diabetes, clearly qualifying these diseases as senopathies.

## 4. Conclusions

While we are moving towards precision medicine with more disease entities identified by molecular patterning than before by clinical phenotypic classification alone [84], one common denominator of several very different and heterogeneous conditions emerges: cellular senescence is present and accumulates in a plethora of mainly, but not exclusively, age-associated diseases and can be targeted by senotherapies as a single treatment strategy. Therefore, we postulate the term “senopathy” to define such diseases and conditions. One caveat, however, is the so far not well-described and understood heterogeneity of senescence depending on different senescence triggers and specific cell types. At this point, developing single senolytic or senomorphic drugs that specifically target all senescent cells in an organism is not well-conceivable.

## Figures and Tables

**Table 1 biomolecules-13-00966-t001:** Examples of medical conditions with evidence for an implication of senescent cells. Italic font indicates mRNA. “Age” reflects the age at analysis. * Initial age of mice before the start of the treatment is not specified; mo: month; WT: wildtype; p16-3MR: p16^INK4A^ trimodality reporter; HFD: high-fat diet; INK-ATTAC: p16^INK4A^-induced apoptosis through targeted activation of caspase; SAMP8: Senescence Accelerated Mouse-Prone 8; PS19: Tau P301S (PS19Tg); NOD: Non-Obese Diabetic Mouse; db/db: BKS.Cg-Dock7^m^+/+Lepr^db^J; D+Q: Dasatinib and Quercetin; SASP: senescence-associated secretory phenotype; TAF: telomere-associated DNA damage repair foci; AECII: alveolar epithelial type II cell; SA-β-gal: senescence-associated β-galactosidase activity.

Organ/System/ Diseases	Age/Genotype	Senolytic Treatment	Evidence for Targeting SCs	Effects	Reference
Cardiovascular system	24-mo-old WT	D+Q	SA-β-gal, *p16^INK4A^*	Improved cardiovascular function	[16]: Zhu et al., 2015
	23-mo-old WT	navitoclax	TAF, *p16^INK4A^*	Reduced hypertrophy and fibrosis, increased regeneration	[62]: Anderson et al., 2019
	8-mo-old p16-3MR; Ldlr^−/−^ HFD	navitoclax	SA-β-gal, *p16^INK4A^*, *pl9^ARF^*, *p21^WAF1/CIP1^*, *IL-1a*, *Mmp3*, *Mmpl3*	Reduced plaque number and size, smaller fatty streaks, and protection from plaques rupture	[63]: Childs et al., 2016
Fat tissue	24-mo-old WT	ruxolitinib	SA-β-gal, *IL-6*, *p21^WAF1/CIP1^*, *p16^INK4A^*	Enhanced adipogenesis, reduced loss of fat tissue, reduced lipotoxicity, and enhanced insulin sensitivity	[64]: Xu et al., 2015
	min. 4-mo-old * INK-ATTAC HFD	D+Q	SA-β-gal, *p16^INK4A^*, TAF, SASP factors in plasma	Alleviated anxiety-related behavior and decreased inflammation	[66]: Ogrodnik et al., 2019
Bones and cartilages	20-mo-old WT	D+Q	*p16^INK4A^*, bone fluorescence, distention of satellites osteocytes	Higher bone mass and strength, better bone microarchitecture, reduced osteoporosis, and frailty	[71]: Farr et al., 2017
	22-mo-old WT	ruxolitinib			
	19-mo-old p16-3MR	UBX0101	*p16^INK4A^*, *p21^WAF1/CIP1^*, *11-6*, *Mmpl3*	Inhibited articular cartilage erosion	[72]: Jeon et al., 2017
	5-mo-old Zmpste24^−/−^	SR12343	SA-β-gal, *p16^INK4A^*, *p21^WAF1/CIP1^*, *IL-6*	Inhibited fibrosis formation (trichrome staining), improved myogenesis	[75]: Zhang et al., 2021
Liver and kidney	4-mo-old Mdr^−/−^	A-1331852	*p16^INK4A^*-positive cholangiocytes and SASP levels	Reduction in liver fibrosis and amounts of growth factors and cytokines	[77]: Moncsek et al., 2018
	24-mo-old INK-ATTAC	D+Q	Karyomegalic and TAF-positive hepatocytes	Reduced hepatic steatosis	[76]: Ogrodnik et al., 2017
	24-28-mo-old p16-3MR	FOXO4-DRI	Lamin B1-positive cells, *IL-6*	Improved kidney function	[21]: Baar et al., 2017
Brain and neurodegeneration	10-mo-old SAMP8	fisetin		Reduced cognitive deficits, restored synaptic function, stress, and inflammation	[78]: Currais et al., 2018
	23-mo-old tau_NFT_-Mapt^0/0^	D+Q	*p16^INK4A^*, *p21^WAF1/CIP^*, *Cxcl1*, *IL-1a*	Neuroprotective effects, improved brain structure, and cerebral blood flow	[79]: Musi et al., 2018
	6-8-mo-old INK-ATTAC; PS19	navitoclax	X-Gal crystals	Prevented gliosis and hyper-phosphorylation of tau, preserved cognitive function	[80]: Bussian et al., 2018
Fibrotic pulmonary disease	4-9-mo-old INK-ATTAC bleomycin-treated	D+Q	*p16^INK4A^*, *Mcpl*, *11-6*, *Mmpl2*	Improvement of lung function measured by plethysmography, increased body weight, and improved exercise capacity	[81]: Schafer et al., 2017
	8-9-mo-old WT irradiated	navitoclax	SA-β-gal positive AECIIs, *p16^INK4A^*, *Bcl-2*, *IL-1a*, *IL-1b*	Reversed pulmonary fibrosis	[82]: Pan et al., 2017
Diabetes	9-mo-old NOD	ABT-737 ABT-199	SA-β-gal, *p16^INK4A^*, *p21^WAF1/CIP^*, *Igfbp3*, *IL-6*	Prevention of type-1 diabetes by inhibition of b-cell destruction	[65]: Thompson et al., 2019
	min. 2-mo-old * db/db mice	D+Q	SA-β-gal, *p16^INK4A^*, TAF	Alleviated anxiety-related behavior	[66]: Ogrodnik et al., 2019

## Data Availability

Not applicable.

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
