# Peer review of "Senopathies—Diseases Associated with Cellular Senescence"

_biomolecules, 2023, doi:10.3390/biom13060966_

Round 1

Reviewer 1 Report

The review by  Lushchak address the issue of cellular senescence. It is not a novel issue but I think that this short review could be of interest for the readers since it summarizes the most relevant infomation on senescent cells.

Author Response

We are grateful that this reviewer is very positive on our opinion article.

Reviewer 2 Report

In this manuscript Lushchak et al. present a synthetic revision of pathologies in which senescent cells have been identified to play a role in the pathogenesis or progression of the disease, and refer to them as “senopathies”. This classification is, to my knowledge, new and could help the development or application of senotherapies. Overall, the manuscript is well written and provides an interesting point of view, that merits publication. Below are some suggestions to strengthen and clarify some of the contents of the revision, that should be addressed prior to publication.

1. In line 32, when referring to senomorphics the authors should include that senomorphics reduce or modify the senescence associated secretory phenotype (SASP).

2. Please provide examples and references for of the interesting concept presented on line 33, regarding cellular senescence promoting tissue damage by cell de-differentiation.

3. Including the existing evidence of the impact of senomorphics (compounds targeting specifically the SASP) on the development of diseases would strengthen the manuscript.

4. The section on Neuropathies should be expanded, clarifying to which diseases are the authors referring to.

5. In Table 1, please include the number corresponding to each reference and a list with all the abbreviations.

6. Please include in the text (maybe in sections in lines 122-175) more information on the animal models presented in Table 1 in the column Age/genotype.

Author Response

In this manuscript Lushchak et al. present a synthetic revision of pathologies in which senescent cells have been identified to play a role in the pathogenesis or progression of the disease, and refer to them as “senopathies”. This classification is, to my knowledge, new and could help the development or application of senotherapies. Overall, the manuscript is well written and provides an interesting point of view, that merits publication. Below are some suggestions to strengthen and clarify some of the contents of the revision, that should be addressed prior to publication.

Thank you very much for the positive evaluation of our manuscript and for the constructive suggestions for improvements.

  1. In line 32, when referring to senomorphics the authors should include that senomorphics reduce or modify the senescence associated secretory phenotype (SASP).

We changed the respective passage as follows:

These discoveries initiated ongoing endeavors to identify novel „senotherapies“, which comprise pharmacological compounds to eliminate SCs specifically („senolytics“) or to mitigate or modify their senescence-associated secretory phenotype (SASP)  („senomorphics“).

  1. Please provide examples and references for of the interesting concept presented on line 33, regarding cellular senescence promoting tissue damage by cell de-differentiation.

We inserted the following paragraph (including references) at line 44:

Cellular senescence is also associated with increased cell plasticity and a concomitant loss of cell identity. Senescent cells promote the de-differentiation of non-senescent neighboring cells, causing further tissue damage [29]. This relatively new concept is supported by a study demonstrating that Alzheimer's disease (AD) patients' induced neurons simultaneously expressed markers of de-differentiation and cellular senescence [30].

  1. Including the existing evidence of the impact of senomorphics (compounds targeting specifically the SASP) on the development of diseases would strengthen the manuscript.

The following new paragraph at line 121 generally discusses current evidence for the impact of senomorphics on senopathies:

Similar mechanistic relationships are more difficult to establish for senomorphics because these compounds target the SASP, not SCs as such. Since most SASP components are commonly present in inflammatory diseases, it is difficult to determine if a therapeutically effective compound's mode of action is via its senomorphic or more general anti-inflammatory properties. Thus, readouts, such as the most common SASP factors IL-6 or IL-8, are insufficient to establish a clear mechanistic connection between SCs and a specific disease. Still, several compounds with senomorphic activity in vitro, such as rapamycin, metformin, aspirin, statins and NF-κB-, p38MAPK-, JAK/STAT-, and ATM-inhibitors, are effective in potential senopathies [63].

We now also included metformin in the chapter on diabetes (line 155):

Metformin is an approved drug for type 2 diabetes in humans, but also positively affects other age-related disorders [69] and extends the healthy lifespan of male mice [70]. Although several senescence markers in different cell types are down-regulated upon exposure to this drug [71,72], the precise mechanisms of metformin action remain elusive [63].

and discuss aspirin and SR12343 the the context of muscoskeletal diseases (line 164):

Aspirin, for which senomorphic properties were described [75], is also effective in arthritis and osteoporosis [76]. The novel small molecule NF-κB-inhibitor SR12343 reduced senescence markers and improved muscle pathologies in the Zmpste24−/− progeroid mouse model [77].

  1. The section on Neuropathies should be expanded, clarifying to which diseases are the authors referring to.

We modified the chapter on neurodegeneration (line 177) accordingly:

One prominent example of a neurodegenerative condition likely promoted by the accumulation of SCs and characterized by the expression of tau is AD. Studies in SAMP8 mice developing pathophysiological alterations similar to human AD and in the tauNFT‐Mapt0/0 model acquiring tau-related pathologies at a faster rate show that Ttreatments with fisetin or D+Q decreased inflammation and stress with restored synaptic function, cognitive deficits, and cerebral blood flow [80,81]. Moreover, navitoclax prevented cognitive function decline, gliosis, and hyper-phosphorylation of tau in INK-ATTAC PS19 mice, which express high levels of human tau, specifically in neurons  [82]. 

  1. In Table 1, please include the number corresponding to each reference and a list with all the abbreviations.

We changed Table 1 as suggested and inserted an expanded figure legend explaining all abbreviations.

  1. Please include in the text (maybe in sections in lines 122-175) more information on the animal models presented in Table 1 in the column Age/genotype.

We now briefly explain the INK-ATTAC and p16-3MR models in line 133:

Several studies used the p16-3MR [32] or the INK-ATTAC [14] mouse model. Both models contain a transgene that permits the visualization and inducible elimination of p16INK4A-expressing cells.

Moreover, we included more comprehensive information on the different disease models in the sections of the respective pathologies (line 136 – 183).

Reviewer 3 Report

Lushchak et al. effectively explains the senescence phenotype and its impact on aging and age-associated diseases. They have described the heterogeneity of the senescent cells, and the conclusion about not getting a single senolytic target is truly appropriate.

Author Response

We are happy to hear that there are no revisions necessary and the reviewer is very positive about our manuscript.